# An Analysis Scheme of Balancing Energy Consumption with Mobile Velocity Control Strategy for Wireless Rechargeable Sensor Networks

**DOI:** 10.3390/s20164494

**Published:** 2020-08-11

**Authors:** Shun-Miao Zhang, Sheng-Bo Gao, Thi-Kien Dao, De-Gen Huang, Jin Wang, Hong-Wei Yao, Osama Alfarraj, Amr Tolba

**Affiliations:** 1School of Computer Science and Technology, Dalian University of Technology, Dalian 116024, China; zshunmiao@mail.dlut.edu.cn; 2Fujian Provincial Key Laboratory of Big Data Mining and Applications, Fujian University of Technology, Fuzhou 350118, China; Gshengbo@smail.fjut.edu.cn (S.-B.G.); 1101405123@nkust.edu.tw (T.-K.D.); jinwang@csust.edu.cn (J.W.); Yhongwei@smail.fjut.edu.cn (H.-W.Y.); 3School of Computer & Communication Engineering, Changsha University of Science & Technology, Changsha 410000, China; 4Computer Science Department, Community College, King Saud University, Riyadh 11437, Saudi Arabia; oalfarraj@ksu.edu.sa (O.A.); atolba@ksu.edu.sa (A.T.); 5Mathematics and Computer Science Department, Faculty of Science, Menoufia University, Shebin-El-Kom 32511, Egypt

**Keywords:** wireless rechargeable sensor networks, mobile gathering and charging, speed control

## Abstract

Wireless Rechargeable Sensor Networks (WRSN) are not yet fully functional and robust due to the fact that their setting parameters assume fixed control velocity and location. This study proposes a novel scheme of the WRSN with mobile sink (MS) velocity control strategies for charging nodes and collecting its data in WRSN. Strip space of the deployed network area is divided into sub-locations for variant corresponding velocities based on nodes energy expenditure demands. The points of consumed energy bottleneck nodes in sub-locations are determined based on gathering data of residual energy and expenditure of nodes. A minimum reliable energy balanced spanning tree is constructed based on data collection to optimize the data transmission paths, balance energy consumption, and reduce data loss during transmission. Experimental results are compared with the other methods in the literature that show that the proposed scheme offers a more effective alternative in reducing the network packet loss rate, balancing the nodes’ energy consumption, and charging capacity of the nodes than the competitors.

## 1. Introduction

Wireless sensor networks (WSN) have been widely applied in various fields [1,2,3], e.g., monitoring power pipelines, subsea tunnels, bridges, smart homes, medicine, and environment [4,5,6]. However, a large number of equipped battery nodes would cause a network to be lost or not work properly whenever its nodes’ batteries are depleted [7,8]. A wireless charging battery is one of the promising ways to have a solution to extend the life of sensor networks [9,10]. A wireless charging battery uses a mobile sink (MS) to charge the sensor nodes to extend the service life of WSNs [11]. It is a combination of WSNs and wireless charging technology known as Wireless Rechargeable Sensor Networks (WRSN). There are some changing types in WRSN, such as single-point and multi-point charging methods [12]. In the single-point charging method, MS can only charge one sensor node by periodically visiting and charging each node to keep the network working regularly [13]. On the other hand, in the multi-point charging method, the MS can simultaneously charge multiple sensor nodes within the charging range [14]. Different from the single-point charging, the MS needs to select the charging stay point or “anchor point” and design the charging path such that the MS charges the nodes within the node range after staying at the staying point. Both data collection, e.g., the position, the strong signal, residual energy, and charging nodes are considered. WRSN uses the mobile charging car for the sensors to avoid dying nodes [15,16]. The other method considered the trade-off relationship between the node charge acquisition amount and the node transmission consumption based on resource allocation to improve the energy efficiency of the system [17]. 

However, most of the mentioned charging methods apply to the fixed traveling velocity of MS as a constant that made the MS spend more time on the trajectory [9]. Especially in a narrow and long space, a running MS may overflow the charge of some nodes, causing an unbalanced phenomenon and an inefficient charger. The nodes in WRSN have high energy consumption, generally because of the need to forward a large amount of data to other nodes. Alternatively, the distance of transmitted data packets is relatively long, which requires more energy from charging capacity than the other nodes with fewer load tasks and low energy consumption [18].

Moreover, there are a few previous works that also considered changing the driving velocity speed of the mobile charger to get better charging ability from the portable charger [19,20]. The distances between the charging points and the residence time of the charging points were regarded to adjust the moving speed on the path and reduce the charging completion time. These methods have the advantages of determining the optimal rate of the mobile data receiver and avoiding the energy hole problem by analyzing transmission energy consumption.

However, the bottlenecks points of node energy consumption and the data transmission paths optimization have been not considered seriously for the core issue of how to control the charger’s moving speed. Therefore, the energy consumption of the cluster head nodes (whose load is much higher than other nodes) would die faster, which will lead to the performance degradation of the entire network.

This paper considers adjusting the velocity of the moving MS based on optimization of the requirement of the nodes charging energy amount and joint data collection in order to maintain the regular operation of the network. The highlights of the contribution in this paper are listed as follows.
Analyzing the MS velocity speed control problem in the WRSN by determining the points of the bottlenecks’ node energy consumption with higher energy consumption nodes in WSN and the energy consumption of the bottleneck nodes.Optimizing the data transmission paths by constructing reliable energy balanced spanning trees based on data collection and residual energy network.Suggesting a moving speed control scheme for MS collecting data and charging nodes in a strip-based area according to the optimal demand of the node to extend the life of the network.Implementing many experiments to verify the reliability of the proposed scheme for the MS speed control through a two-distinction routing graph for charging and collecting data to avoid hot spots and reduce the data packet loss rate.

The rest of the paper is organized as follows. Section 2 discusses related work with a system model and problem statement. Section 3 presents MS velocity control strategies for optimizing energy when designing WRNS. Section 4 gives the experimental results and discussion. Section 5 summarizes the work.

## 2. System Model and Problem Statement

This section reviews the system model of WRSNs [14,15] and describes the problem statement as follows.

### 2.1. System Model

Assume a deploying network with randomly distributed N wireless rechargeable sensors (S = {$S_{1},\;\;S_{2},\;\;S_{3},\;\ldots ,\;S_{N}$}) in a strip space of length L and width W, where W<<L, for collecting ambient data. A mobile sink (MS) travels on a pre-designed trajectory at different speeds *V*, receives the data transmitted by the sensor, and charges all nodes within the charging range of wireless charging transmission power. The MS is undertaken moving along a predefined linear trajectory in a two-dimensional space that is not affected by external environmental factors. While traveling, the MS also can collect data from sent sensor nodes, and charge the sensor nodes in transmission range. This means that an MS is used to charge the sensor nodes and collect data in the strip-based space. The node-aware data needs to be transmitted to the receiver through one or more hops in WSN due to the limited buffer space and energy of each sensor [11]. Figure 1 shows a description of a system model of WRSNs.

The initial energy of each sensor is $E_{full}$ J, and the maximum buffer space is $B_{full}$ KB. In the network, the sensor collects data at *b* bits per second and transmits the data to the receiver according to a specific path. In this process, the relay point transmission is used to send the node data to the MS in a multi-hop manner [21]. We assume that any node within the boundary is a relay node, denoted as node $R_{j}$, and can directly transmit data to MS. Any nodes outside the dividing line are common nodes, denoted as node $C_{i}$, then the data needs to be transmitted through the relay node to the MS.

An energy model is used as a consumption model with the primary energy consumption of nodes in the energy consumed by processing, transmitting, and receiving data [16,22]. Assume that the size of each data packet is k bits. Assume a node $S_{i}$ sends *k* bits to node $S_{j}$, and $d_{ij}$ is expressed as the Euclidean distance of $S_{i}$ and $S_{j}$. The energy consumption of node $S_{i}$ sending a data packet (*k* bits) can be written as follows:(1)$$E_{ij}^{tr}=\{\begin{array}{c} k\left(E_{elec}+E_{fs}d_{ij}^{2}\right)\;,d_{ij}\leq d_{m}\\ k\left(E_{elec}+E_{mp}d_{ij}^{4}\right)\;,d_{ij}>d_{m} \end{array}$$

The energy consumed by each node receiving one data packet (*k* bits) is expressed as:(2)$$E_{ij}^{re}=kE_{elec}$$
where $E_{elec}$ is the electronics energy, $E_{fs}$ and $E_{mp}$ are the amplifier energy of the free space and multi-path models, $E_{elec}$ = 50 nJ/bit, $E_{fs}$ = 10 pJ/bit/m^2^, $E_{mp}$ = 1.3 × 10^−3^pJ/bit/m^4^; $d_{m}$ = 87 m [23]. The energy consumption model of a node is an assumed, free space model, that is, $d_{ij}$ < $d_{m}$; *k* bits refers to the amount of data transmitted by the node and the amount of data received by the node [24]. 

$B_{k}\left(T\right)$ refers to the buffer space capacity of the sensor *S_k_* at time T. Based on the data transmission-reception model, the calculation expression for the buffer space of the sensor node *S_k_* can be given as follows:(3)$$B_{k}\left(T\right)=\{\begin{array}{c} {\left[B_{k}\left(T-1\right)-b_{kd}^{tr}+b_{k}\right]}^{+}\;\;\;\;\;\;\;,k\in C,d\in R\;\;\;\;\;\\ {\left[B_{k}\left(T-1\right)-b_{kd}^{tr}+b_{k}+b_{dk}^{re}\;\right]}^{+}\;,k\in R,d\in C,s\in S \end{array}$$
where $b_{k}$ refers to the rate at which the sensor node *S_k_* collects the amount of data, *b* bit/s; $b_{kd}^{tr}$ refers to the amount of $b_{kd}$ bit data transmitted from the sensor node *S_k_* to the sensor node *d*; $b_{dk}^{re}$ refers to the sensor node *S_k_* receiving the $b_{kd}$ bit data transmitted from the sensor node *d* the amount. $B_{full}$ is the maximum cache space capacity of the sensor node, ${\left[Q\right]}^{+}=\;\min\left(\mathrm{FULL},\;\max\left(0,Q\right)\right)$. 

The wireless charging model in [19] is described as follows: the MS has a circle charging range with a radius of R (in the experiment, R is set to 30 m). The sensor nodes have a rechargeable receiver, and the received power of the node is determined by distance and the transmission power. The charging model is expressed as follows:(4)$$P_{kr}\left(d\right)=\frac{G_{s}G_{r}\eta P_{t}}{L_{p}}{\left(\frac{\lambda }{4\pi \left(d_{k}+\beta \right)}\right)}^{2}$$
where ${\;P}_{kr}$ is the received power of node *S_k_*, $d_{ks}$ is the Euclidean distance between the MS and node *S_k_*, $G_{s}$ represents the antenna gain of the transmitter, $G_{r}$ is the antenna gain of the receiver, $P_{t}$ is the transmitted power of the MS, *η* is the rectifier efficiency, λ represent the signal wavelength, $L_{p}$ is the polarization loss, *β* is a parameter used to adjust the Friis’ free space equation for short-distance transmissions (the experimental values of *β* is 0.2316). Equation (4) can be simplified as follows:(5)$$P_{kr}\left(d\right)=\frac{\omega }{{\left(d_{ks}+\beta \right)}^{2}}$$
(6)$$\omega =\frac{G_{s}G_{r}\eta P_{t}}{L_{p}}{\left(\frac{\lambda }{4\pi }\right)}^{2}$$

In the charging model, the node’s received power is inversely proportional to the square of the distance between the node and the MS. As the distance increases, the node’s charging power will decrease sharply. We assume that the moving distance of the MS in the *T* period is *S(T)*, *S(T) = V*T*, so the distance $d_{ks}$ between the node *S_k_* and the MS is written as follows:(7)$$d_{ks}=\int_{0}^{T}\sqrt{{\left(y_{k}-y_{s}\right)}^{2}+{\left(x_{k}-x_{s}^{ts}-V\cdot t\right)}^{2}}dt$$
where $\left(x_{k},y_{k}\right)$ refers to the coordinates of the node *S_k_*; $y_{s}$ refers to the y-axis coordinates of the MS moving trajectory; $x_{s}^{ts}$ refers to the *x*-axis coordinate when the MS can charge the node *S_k_*. This coordinate can be obtained from the coordinates of the charging range $d_{c}$ and the node *S_k_*. The calculation formula is as follows:(8)$$x_{x}^{ts}=max\left(x_{k}-\sqrt{d_{c}^{2}-{\left(y_{k}-y_{s}\right)}^{2}},0\right)$$

### 2.2. Problem Statement

In the entire strip-shaped WSN, the sensor battery-equipped nodes are randomly deployed and organized into a running network [25,26], and its nodes are exhausted of energy [27]. The system works appropriately and has a long life. Its sensor battery-equipped nodes need to be charged to add power. However, directly charging the nodes is an impossible task because of the large number of nodes and deployed hardware places. As mentioned, mobile charging is a promising way to deal with this issue. Controlling the moving speed of MS will help the charging time of the nodes by balancing energy for all nodes in maintaining the life cycle of the network. The service life of the system can be converted into the speed control sub-problem and the data transmission path selection sub-problem in the MS mobile charging process.

In order to maximize the life cycle of the sensor network, two questions are answered. How can we control the moving speed of the MS? How can we efficiently transfer data to the MS in the speed control strategy? This statement can be formulated objective function *f* as follows:(9)$$\begin{array}{cc} & f=\max\frac{\underset{i}{\min}(E_{i}^{charge}-E_{i}^{consumption})}{average(\sum^{\;}E_{i}^{charge}-E_{i}^{consumption})},\;i\in K\\ & s.t.\;C1:\;E_{k}^{charge}>E_{k}^{consumption}\\ & C2:\;E_{k}^{consumption}\leq E_{k}\\ & C3:\;E_{k}\leq E_{full}\;or\;E_{k}\leq E_{k}^{helthy}\\ & C4:\;B_{k}\leq B_{full}\;or\;B_{k}\leq B_{k}^{helthy} \end{array}$$

When the target value of *f* is more significant, it means that the charging amount of the smallest charging node is closer to the average cost, and the charging is more balanced. The constraint of C1 ensures that all sensor nodes can be charged, and the charging amount is higher than the node consumption. C2 ensures that all nodes will not die before charging, to avoid the phenomenon of moving too fast or too slow during speed control. The constraints C3 and C4 ensure that at the end of charging, the node reaches a fully charged state or charging threshold, and the buffer space will not overflow. 

When calculating node energy consumption and charging acquisition, we replace the time-varying distance between the node and the MS by estimating the fixed transmission and charging distance, which makes it easier to control the moving speed. Table 1 lists the commonly used symbols and definitions in this paper.

## 3. MS Velocity Control Strategies for Optimizing Energy

This section presents the control strategies of MS moving speed through the system analyzation and optimization paths for MS traveling. The presentation includes subsections, e.g., system analysis, speed control strategies, optimizing energy consumption, and best path selection with a minimum spanning tree for reliable balance energy.

### 3.1. System Analysis

Refer to Equation (9), if MS moves slowly (denoted V), the charged node’s amount ($E_{s}^{charge})$ will be obtained more fully [28]. Since the charging demand of each sensor node is different, and the charging interval may overlap, the moving speed of the MS needs to be controlled in the node charging interval. Figure 2 shows a strip space with different regions to determine the charging areas. Assume each velocity zone is determined by the least residual energy of nodes, for which reason this node is the “bottleneck node” in this interval. The strip space can be divided into a plurality of different speed regions to determine the moving speed of MS. In the split interval, if there is one node Si, it is only necessary to satisfy the charging demand of the node Si, so that its actual charging is maximized, and the charging state is the charging threshold at the end of charging. Correspondingly, other nodes can also meet the charging needs. The charging bottleneck nodes meet the requirements of the objective function, so the bottleneck node can be used to determine the speed V of each stage of the MS. The entire network MS moving speed set $V=\left\{v_{1},\;\;v_{2},\;\;v_{3}\ldots \right\}$ to complete the speed control strategy. 

**Theorem** **1.**
*For all sensor nodes within the MS charging range, the required time for node charging is set to $T_{c}$, $T_{c}=\left\{t_{1},\;\;t_{2},\;\;t_{3},\ldots \right\}$. If all nodes are charged with the charging time $t_{Si}$ of node S_i_, and $t_{Si}$ is the longest time, $i.e.,\;\;t_{Si}=\max T_{c}$; then, all sensor nodes can eventually reach a fully charged state.*


**Proof.** It is assumed that all nodes need to transmit the data with the maximum packet size of M bits to the MS while charging. According to the node transmission distance $d_{k}$ and charging time $t_{k}$, we have the actual charging of the node as follows:(10)$$E_{k}^{practical}=\int_{0}^{t_{k}}\frac{\omega }{{\left(d_{k}+\beta \right)}^{2}}dt-M\times \left(E_{elec}+E_{fs}\times d_{k}\right)$$Putting $T_{c}$ into Formula (10), we can obtain
$E_{i}=\int_{0}^{t_{i}}\frac{\omega }{{\left(d_{i}+\beta \right)}^{2}}dt-M\times \left(E_{elec}+E_{fs}\times d_{i}\right),t_{i}\in T_{c},i=1,2,3,\cdots K,$
Among them, there is a node *S_i_*,$t_{Si}=\max T_{C}$. Substituting $t_{Si}$ for $T_{c}$, we can obtain
$$E_{i}^{\prime }=\int_{0}^{t_{Si}}\frac{\omega }{{\left(d_{i}+\beta \right)}^{2}}dt-M\times \left(E_{elec}+E_{fs}\times d_{i}\right)\geq E_{i},i=1,2,3,\cdots K,$$Then the actual charging capacity of each node will be higher than the original charging capacity. Therefore, when meeting the charging demand of node *S_i_*, other nodes can also meet the requirements. We call this kind of node *S_i_* a “bottleneck node”. □

Figure 3 shows the type of moving speed of MS in continuous and converted into a discrete one. In the WRSNs, speed control is a constant process (Figure 3a), and continuous change is more complicated, so we discretize the speed (Figure 3b). A centralized algorithm is proposed for charging requirements to control MS speed based on the determined bottleneck nodes. In the entire strip space, there may be multiple bottleneck nodes that area would be divided into several sub-bottleneck regions according to the charging interval of the bottleneck nodes.

As mentioned in Section 2, the energy consumption of nodes and the charging amount required for nodes would help to find the charging time for moving MC. The speed control strategy for MS depends on the nodes’ location, residue energy, bottleneck area, and moving trajectory of MC. 

The determining bottleneck nodes have to consider the same initial condition and capacity of the nodes $B^{init}$ and the initial energy $E^{init}$. Then the energy consumption $E_{i}^{consumption}$ of the initial transmission of the node, the charging gain $E_{kh}^{\prime }$ per unit time, and the charging interval is calculated. For the common node *S_i_*, it needs to indirectly transmit the collected data to the MS through the relay node *S_j_*. The energy consumption formula $E_{i}^{consumption}$ of the node *S_i_* transmitting to the relay node *S_j_* is as follows:(11)$$E_{i}^{consumption}=B_{i}^{init}\times \left(E_{elec}+E_{fs}\times d_{ij}^{2}\right)$$

Among them, $B^{init}$ represents the initial cache of the node.
(12)$$d_{ij}=\sqrt{{\left(x_{i}-x_{j}\right)}^{2}+{\left(y_{i}-y_{j}\right)}^{2}}$$

The common node *S_i_* is far away from the MS; if its data is directly transmitted to the MS, it will consume more energy. The relay node indirect transmission method can effectively reduce the energy consumption of the common node *S_i_*. However, if the relay node *S_j_* has a large number of common nodes within one hop and sends all data to the relay node, the burden on the relay node *S_j_* is increased. 

The energy consumption of the relay node is mainly divided into two parts: transmission consumption and reception consumption. The transmission range of the node is limited; the distance between the MS and the node is less than the transmission range, the relay node will directly send data to the MS. When the MS does not move to the node’s transmission area, the node stores the data collected by itself and the received data in the buffer space first. $G_{j}$ denoted the number of common nodes that need to be forwarded by the relay node *S_j_*, $G_{j}$ is determined by the buffer space and the remaining energy of the relay node. The energy consumption formula of relay node *S_j_* is as follows:(13)$$E_{j}^{consumption}=\left(B_{j}^{init}+\overset{G_{j}}{\underset{i=1}{\sum }}B_{i}^{tr}\right)\times \left(E_{elec}+E_{fs}\times d_{js}^{2}\right)+G_{j}\times B_{j}^{re}\times E_{elec}$$

Among them, $B_{j}^{re}$ represents the data received by the relay node *S_j_*, $B_{j}^{re}=\sum^{\;}B_{i}^{tr}$. Without considering the amount of data received, the energy consumption of the relay node in Equation (13) is only related to the transmission distance. The relay node is directly sent to the MS through a single hop, so the transmission distance $d_{js}$ of the relay node will change due to the change of the MS position. 

For a more intuitive analysis, we take the coordinates $\left(x_{j},y_{j}\right)$ of the node *Sj* as the center *O*, and the transmission radius $d_{tr}$ as the circle radius. As shown in Figure 4, $L_{tr}$ represents the transmission interval. It is assumed that sink moves along a fixed straight path with $y=y_{s}$ [18]. Thus, the corresponding formula for the transmission distance of node j is as follows:(14)$$d_{ij}=\sqrt{h_{j}^{2}+{\left(x_{j}-V\times t\right)}^{2}}\;t\in \left(t_{j}^{ts},t_{j}^{te}\right)$$
where the time use $t_{j}^{ts}\;and\;t_{j}^{te}$ when the first MS enters the transmission range of the node *S_j_* and the time when it leaves the transmission range.

Assuming that the MS sends M bit data every time when the node *S_j_* transmits the interval and does not consider the reception consumption, the total consumption of the node *S_j_* is written as
(15)$$E_{j}^{total}=\int_{t_{j}^{ts}}^{t_{j}^{te}}M\times \left(E_{elec}+E_{fs}\times d_{js}^{2}\right)dt$$
which can be converted to
(16)$$E_{j}^{total}=M\times E_{elec}\times \left(t_{j}^{te}-t_{j}^{ts}\right)+M\times E_{fs}\times \int_{t_{j}^{ts}}^{t_{j}^{te}}d_{js}^{2}dt$$

According to Equation (14), the distance $d_{js}$ between the relay node and the MS changes with time. In practice, the time to transmit data is much shorter than the MS’s travel time, which is not conducive to the energy consumption of the computing node. Therefore, Equation (16) can be simplified, and set the transmission distance at each moment as a fixed value $d_{js}^{\prime }$, and the fixed value $d_{js}^{\prime }$ takes the middle of the transmission distance, $d_{js}^{\prime }=\left(d_{js}^{max}+d_{js}^{min}\right)/2$, $d_{js}^{max}$ and $d_{js}^{min}$ respectively represent the furthest distance and shortest distance from the relay node to the MS within the data transmission range of the node. Replacing calculations with fixed values will increase the total consumption of node calculations. If the speed control can be completed under the condition of energy consumption amplification, the speed control is also feasible in actual situations. The simplified formula is as follows:(17)$$E_{j}^{total}=M\times \left(t_{j}^{te}-t_{j}^{ts}\right)\times (E_{elec}+E_{fs}\times d_{js}^{\prime 2})$$

For the sensor node *S_k_*∈*S*, we refer to a fixed value $d_{ks}^{\prime }$ instead of the distance between the node *S_k_* and the MS. We make the fixed value $d_{ks}^{\prime }=\left(d_{ks}^{max}+d_{ks}^{min}\right)/2$, and $\left(t_{j}^{te}-t_{j}^{ts}\right)\times {\left(d_{js}^{max}+d_{js}^{min}\right)}^{2}/4\geq \int_{t_{j}^{ts}}^{t_{j}^{te}}d_{js}^{2}dt$, substituting $d_{js}^{\prime }$ into Equation (17) gives.
(18)$$E_{j}^{total}=M\times \left(t_{j}^{te}-t_{j}^{ts}\right)\times \left(E_{elec}+E_{fs}\times {\left(\frac{d_{js}^{max}+d_{js}^{min}}{2}\right)}^{2}\right)$$

In terms of mobile charging, MS is a wireless charging source and uses mobile charging. The amount of charge for the node is determined by the charging distance and charging time. Therefore, we can get the formula for obtaining the charging energy of the sensor node *S_k_*.
(19)$$E_{k}^{h}=\int_{t_{k}^{cs}}^{t_{k}^{ce}}\frac{\omega }{{\left(d_{k}+\beta \right)}^{2}}dt$$
where $t_{k}^{cs}\;and\;t_{k}^{ce}$ represent the time points when the sensor node *S_k_* enters and leaves the MS charging range.

For each sensor node, there is also a charging interval similar to the transmission interval. The charging range can be used, as shown in Figure 4. The charging distance of the sensor node *S_k_* is calculated according to Equation (14), $t\in \left(t_{k}^{cs},t_{k}^{ce}\right)$. We simplify the charging distance according to Equation (17). The simplified Formula (19) can be written as
(20)$$E_{k}^{h}=\frac{\omega }{{\left(d_{k}^{\prime }+\beta \right)}^{2}}\times \left(t_{k}^{cs}-t_{k}^{ce}\right)=E_{kh}^{\prime }\times T$$
where $\;E_{kh}^{\prime }$ represents the energy obtained by charging the node *S_k_* per unit time. The result of Equation (20) is to reduce the actual charge of node *S_k_* in the calculation. Therefore, according to Equation (20), we can redefine the charging model. For the sensor node *S_k_*, the charging amount of the sensor node *S_k_* is calculated from the energy $E_{kh}^{\prime }$ obtained in the unit time of the node *S_k_* and the time T the MS travels in the charging interval of the node *S_k_*.

According to the energy consumption Equation (20) and charging acquisition [29], the derivation expression of the battery capacity of the sensor node *S_k_* at time T.
(21)$$E_{k}\left(T\right)={\left[E_{k}\left(T-1\right)-E_{k}^{consumption}+E_{k}^{charge}\right]}^{+}$$
where $E^{consumption}$ refers to the transmission consumption of the sensor node, and the energy consumption of the common node and the relay node is obtained by Equation (11) or Equation (13) respectively; $E^{\mathrm{charge}}$ refers to the amount of charging of the sensor node, $E^{charge}=E^{h}$.

### 3.2. Speed Control Strategies

The task is to find a method to ascertain the appropriate moving speed of the MS after determining the energy consumption and charging of the node. The strategy of the appropriate moving speed of the MS is figured out based on dividing into multiple speed areas according to the charging interval of the bottleneck nodes. This method is the discrete rate of searching for bottleneck nodes and the moving trajectory. The rate that MS travels in each area remains constant, as shown in Figure 5. Each area $L_{m}$ corresponds to an area speed $V_{m}$, and the MS moves on the trajectory according to the area speed $V_{m}$, which constitutes a change in the MS moving speed. The time required to charge each node is calculated based on the energy consumption of the node and the charging per unit time, thereby filtering out the bottleneck nodes. The time needed for charging is the time when the sensor node is fully charged, and it can also represent the time that the MS needs to travel in the charging zone of the sensor node *S_k_*. Its expression is written as follows:(22)$$T_{kc}=\left(E_{full}-E_{k}^{pre}\right)/E_{kh}^{\prime }$$
where $E^{pre}$ is the current energy state of the node, which derived from Equation (21). 

For each node, after obtaining the moving time of the MS in the charging interval, the speed of the MS in the node charging interval $L_{m}$ can be obtained according to the charging range of the node.
(23)$$v_{m}=L_{m}/T_{mc}$$

The charging interval of the node is the driving interval of the car that the MS can charge the node, and the driving speed $v_{m}$ in each driving interval [30]. Figure 5 shows multiple areas divided on the basis of speed are obtained, and the node corresponding to the minimum speed is the bottleneck node. According to Theorem 1, in these bottleneck nodes, the MS the speed is controlled as discretization of $V=\left\{v_{1},v_{2},v_{3},\ldots \right\}$. Once the entire calculated speed control is estimated by a fixed value, the complicated computational time (as in Equation (22)) will be increased exponentially. The speed control strategy provides the feasibility of the practice changes to reduce complexity, and thereby the charging process is implemented smoothly. The pseudo-code of the MS speed control algorithm with selected bottleneck points is expressed as follows (Algorithm 1):
**Algorithm 1:** Select bottleneck nodes.**Input:** node *K, L, S***Output:** speed region and bottleneck nodes1:**initialize:**$B^{init}$*,*$E^{init}$2:**for** each *i*∈*K*
**do**3: Classufy *C, R*4:**end for**5:**for** each *i*∈*K*
**do**6: Calculate *the corresponding interval FW_i_*7: **if**
*i*∈*C*
**then**8:  *Select the appropriate relay node*9:  Calculate *node i energy consumption EB_i_,*10: **eles if**
*i*∈*R*11:  *Count the number of nodes C that relay node i need to assist in forwarding datas*12:  Calculate *the corresponding energy consumption EB_i_,*13: **end if**14: Calculate *the amount of charge per unit time EH_i_*15: Update *Ehelthy and Charging requirements*16: Calculate *Charging time TC_i_ and Moving speed VC_i_*17:**end for**18:*Merge interval to divide speed region and select bottleneck node*

The sensor nodes are divided into common nodes and relay nodes with horizontal lines 15 m from both sides of the MS moving trajectory and the dividing line in Algorithm 1 is implemented according to Equations (11)–(23). The transmission interval $L_{tr}$ of each node, the node energy consumption EB and the charging amount EH per unit time are calculated. The charging time TC and the charging moving speed VC is thereby obtained. The transmission interval of the node is recorded as FW. Then, the nodes’ x-axis coordinates are sorted from small to large, and their transmission intervals are concatenated in the sorted order. The MS moving speed is updated according to the spliced speed area to complete the speed control because of the basis of splicing selection is the charging time and moving speed of the node, and the selected node is the bottleneck node.

### 3.3. Optimizing Energy Consumption

The energy consumption of nodes will increase as collecting data in real-time with the increase in time that causes unbalanced energy consumption of the nodes. The power consumption also needs to consider optimizing the speed of the MS through charging requirements.

The sensor node not only causes energy consumption by transmitting data in the initial cache space but also causes energy consumption by continually collecting data during the MS moving process. Nodes farther away from the MS’s starting position have a longer waiting time. The more collected data during the waiting process, the more energy consumption of the node that causes the charging time of the node to increase. This could even lead to the death of the node. Therefore, improving the energy consumption of Equations (11)–(13) can be expressed as follows:(24)$$E_{i}^{consumption}=(B_{i}^{init}+b\times t_{i}^{ts})\times \left(E_{elec}+E_{fs}\times d_{ij}^{2}\right)$$
(25)$$E_{j}^{consumption}=\left(B_{j}^{init}+b\times t_{j}^{ts}+\overset{G_{j}}{\underset{i=1}{\sum }}B_{i}^{tr}\right)\times \left(E_{elec}+E_{fs}\times d_{js}^{2}\right)+G_{j}\times B_{j}^{re}\times E_{elec}$$
where *b* is the sensor acquisition rate, the unit is bit/s. The priority of the charging demand of the node with the least energy would be to set a longer charging time. This means that all nodes within the MS charging range will have extended charging time. Therefore, in terms of energy consumption, buffering, and charging efficiency, a charging optimization scheme is proposed based on a set the energy threshold and the cache threshold to constrain the node’s charging demand and charging time so that the node will not run out of energy. The cache will not overflow before the next MS charge arrives. 

It is necessary to ensure that the energy of the survivable node is above the energy threshold at the end of charging so that the remaining power of the node meets the consumption of data collection and transmission before the next charge of the MS. The limit of the energy threshold can be calculated as follows.
(26)$$E_{k}^{helthy}=\left(G_{k}+1\right)\times \left(T_{s}+t_{k}^{ts}\right)\times b\times \left(E_{elec}+E_{fs}\times d_{k}^{2}\right)+G_{k}\times \left(T_{s}+t_{k}^{ts}\right)\times b\times E_{elec}$$
where $G_{k}$ is the number of nodes assisted by node *S_k_*. When the node *S_k_* is a common node, $G_{k}=0$. $T_{s}$ is the total time of one MS driving. $t_{k}^{ts}$ refers to the time when the MS reaches the transmission range of the sensor node Sk.

The optimized charging demand only needs to keep the energy of the node above the energy threshold. The formula for charging demand is as follows:(27)$$E_{k}^{need}=E_{k}^{helthy}-E_{k}^{pre}$$

Substitute Equation (27) into Equation (22) to optimize the charging time and thus the driving speed.

The number of remaining energy nodes should be determined to support the number of transmitted data packets. According to Equation (25), its relationship can be drawn as follows:(28)$$E_{k}^{pre}=\left(G_{k}+1\right)\times \left(B_{k}^{helthy}+b\times t_{k}^{ts}\right)\times \left(E_{elec}+E_{fs}\times d_{k}^{2}\right)+G_{k}\times \left(B_{k}^{helthy}+b\times t_{k}^{ts}\right)\times E_{elec}$$

The cache threshold should be less than or equal to the amount of data that can be transferred, or the maximum cache space. Thus, according to (28), it can be derived as follows:(29)$$B_{k}^{helthy}\leq \frac{E_{k}^{pre}}{\left(2\times G_{k}\right)\times E_{elec}+\left(G_{k}+1\right)\times E_{fs}\times d_{k}^{2}}-b\times t_{k}^{ts}$$

In order to avoid the death of the nodes and ensure network throughput, it is necessary to combine the energy consumption the optimization formula, energy threshold, and cache threshold, and select bottleneck nodes to optimize the MS moving speed. The pseudo-code of the speed optimization algorithm is seen in Algorithm 2.
**Algorithm 2:** Speed optimization.**Input:** node, *K, L, B, E, v_k_***Output:***V*1:**initialize:***T*2:Calculate *the time before arriving at each node*3:*Transmit data by link*4:Calculate *Ehelthy, Qhelthy, E_need*5:**if***Qk (T) < Qhelthy***then**6: *E_need* = *Ehelthy * θ - E*7:**else**8: *E_need = Ehelthy - E*9:**end if**10:Repeat *Algorithm 1*

In the charging demand calculation of *E_need_*, the energy threshold is amplified by a specific factor to increase node charging demand of more energy to meet the data collection. In the experiment, this factor as coefficient weight θ is set to 1.5.

### 3.4. Best Path Selection 

The relay node transmission method is used to optimize the selection of node transmission paths based on changing the choice of relay nodes, and energy consumption. An energy-balanced spanning tree is constructed to avoid packet loss [31]. Refer to Equation (9), analyzing the energy consumption issues that determine the goal of energy consumption balance.
(30)$$\min\;\frac{\underset{i}{\max}E_{i}^{consumption}}{average\;\sum_{k=1}^{N}E_{k}^{consumption}}$$

The node bidirectional selection data transmission for the data transmission path optimization, e.g., the current node energy, the remaining buffer space, and the charging time are used to obtain the selection relationship between the ordinary node and the relay node. The minimum reliable energy balance spanning tree is constructed through the transmission path of the relay node [32]. The structure of the spanning tree with the weight of the branch contains three factors: transmission distance, residual energy, and transmission energy consumption [33]. A routing cost function is modeled based on the energy consumption part of the weight ($A_{ij}=\mathrm{Link}\_\mathrm{Cos}t()$). Total cost of data transmission in the process of sensor node *S_i_* transmitting data to MS through node *S_j_* can be calculated as follows:(31)$$A_{ij}=B_{i}\times \left(E_{elec}+E_{fs}\times d_{ij}^{2}\right)+\left(B_{i}+B_{j}\right)\times \left(E_{elec}+E_{fs}\times d_{js}^{2}\right)\;i\in C,j\in R$$
where Si is a set of optional relay nodes within a hop range of $\;R_{i}\left\{r_{1},r_{2},\ldots ,r_{j}\right\}$, and $l_{ij}$ is the transmission path link selected for node *i*. Make sure that the buffer space of the relay node does not overflow after receiving the data, and enough energy remains in the node to forward the data. The data transmission chain is reconstructed to reduce the number of branches and energy consumption.

**Definition** **1.***The selection method of the node. For the common node**S_i_**,**S_i_**∈*$C_{i}$*, it starts from*$r_{j}$*in the selectable set*$R_{i}$*until it contains all the select sets*$R_{i}$. *The calculation formula of the path selection weight starting with the relay node**S_j,_**and the selection of a link for the relay*$l_{ij}$*can be written as follows:*(32)$$l_{ij}=\min\left\{\underset{r_{j}}{\sum }\alpha \times \frac{A_{ij}}{\bar{A_{i}}}+\beta \times \frac{\bar{E^{pre}}}{E_{ij}^{pre}}+\gamma \times \frac{Y_{i}}{\bar{Y}}\right\}$$*where*$A_{ij}\;and\;E_{ij}^{pre}$*respectively represent the routing cost of the node**S_i_**selecting the node**S_j_**as a relay and the remaining energy of the relay node**S_j_**;*$\bar{A_{i}}\;and\;\bar{E^{pre}}$*represent the average value of all possible routing costs of node**S_i_**and the average remaining energy of relay nodes, respectively, and*α,β,γ*are weight coefficients, with*α+β+γ=1*.*

Figure 6 shows an example of relay nodes 1, 2, 3 as relays to transmit data to MS, and construct a data transmission link$\left\{R_{a1},R_{a2},R_{a3}\right\}$. The routing costs $A_{a1},A_{a2},A_{a3}$ are calculated according to Formula (30) and substitute them into the routing Formula (31). The weight of each possible path is taken in account for comparison, and the transmission path chain with the smallest weight is selected as the branch. The *R*_a2_ serves as a relay of the ordinary node *Ca*. After all common nodes are selected, *R*_a2_ is selected multiple times. The link reconstruction process can reduce the packet loss rate and the burden on the relay node. Each relay node is only allowed to undertake the forwarding task of a common node and reserve one link. 

A selected set $M_{j}$ is established with the relay node *S_j_* during link reconstruction, where $M_{j}=\left\{m_{1},m_{2},\ldots ,m_{i}\right\}$. All nodes in the set select the node *S_j_* as the relay. The sum of sub-optimal routing costs is computed to ensure that the total routing cost of the network is the lowest. The sum of the sub-optimal routing costs refers to the sum of the optimal-level routing cost $A_{m_{1}}^{opt}$ of the selected node $m_{i}$ and the sub-level routing costs of other candidate nodes $\left\{m_{2},m_{3},\ldots \right\}$, $A_{jm_{1}}^{sub}=A_{m_{1}}^{opt}+A_{m_{2}}^{sec}\ldots +A_{m_{n}}^{sec}.$

**Definition** **2.**
*The selection method of reconstruction. For the relay node*
*S_j_*
*,*
*S_j_*
*∈R, it starts from the node*
$m_{i}$
*in the selected set and ends when all nodes in $M_{j}$ are included. The weight calculation formula for the relay node to select the bearing target is as follows:*
(33)$$m_{ji}=\min\left\{\sum_{m_{i}}\lambda \times \frac{A_{ji}^{sub}}{\bar{A^{sub}}}+\mu \times \frac{\bar{E^{pre}}}{E_{ji}^{pre}}+\nu \times \frac{Y_{j}}{\bar{Y}}\right\}$$
*where*
$m_{ji}$
*indicates that the relay node*
*S_j_*
*selects the common node*
*S_j_*
*as the bearing object.*
$A_{ji}^{sub}$
*represents the sum of sub-optimal routing costs when the relay node*
*S_j_*
*selects the common node*
*S_i_*
*, as in Equation (30). λ,μ,ν are weight coefficients, with λ+μ+ν=1. Figure 7 depicts the ordinary nodes a and b both choose relay node 2 as the relay, and their optimal route costs are*
$A_{a2}^{opt}\;and\;A_{b2}^{opt}$
*, the sub-optimal routing costs are $A_{a1}\;and\;A_{b3}$. The relay node 2 needs to select a bearer object from nodes a and b to perform link reconstruction. The total sub-optimal routing cost of the selected nodes is calculated with $A_{2a}^{sub}$ and $\;A_{2b}^{sub}$, $A_{2b}^{sub}=A_{b2}^{opt}+A_{a1}$. Weight is brought in with the common node corresponding to the smallest weight is selected as the bearer selection result of the relay node 2 to complete the link reconstruction. The flag bit of each relay node is initially set to 0. When the relay node is selected at least twice, its flag bit is set to 1.*


Repeat the comparison and selection until all selections completed. The mutually selected nodes and MS constitutes a reliable and energy-balanced spanning tree. Algorithm 3 describes the main idea of path selection.
**Algorithm 3:** path choice.**Input:** node *K, L, V, B, E***Output:***Link*1:**initialize:***link, list, flag*2:**for** each *i*∈link **do**3: Compute *all the possible value of i as Cost*4: Update *weight*=α**Cost_i_/average_Cost+β*average_E/E_i_+Y_i_/average_Y*5: *Connect i and min weight*6: Update *link*7:**end for**8:**while** list != 0 or flag = 0 **do**9: *Search link for all duplicate nodes as list*10: **for** each *j*∈list **do**11:  Calculate *the second cost of j as Second_Cost*12:  Update *weight* = λ ** Second_Cost_j_ / average_Second_Cost + μ * average_E / E_i_ + ν * Second_Y_i_ / averagr_Second_Y*13:  *Connect i and min weight*14:  Update *link, list, flag*15: **end for**16:**end while**

## 4. Experimental Results and Analysis

Assume a sensor network deployment with N rechargeable sensors (*N* = 100, 200) randomly arranged in the desired area with L×W (*L* = 200, 300; *W* = 50, 100 m) of strip space. The transmission range of the sensor is 20 m. As a mobile receiver and mobile charger [34], MS travels along a fixed trajectory of $y_{s}=L,\;\;x_{s}\in \left(0,\;M\right)$, and its charging range is 30 m. The battery rated capacity of each sensor is 1J, and the cache rated capacity is 500 KB. Collecting the data rate of sensor nodes is set to 512 bits per second; the weight coefficient  α=λ=0.4;β=γ=μ=ν=0.3; and the number of runs conducted is set to 100 for each experiment. The simulation is implemented in Python with the bidirectional routing algorithm. Table 2 lists the simulation parameters setting.

The obtained experimental results of the proposed method (named JGC-MSCS- Joint and Gather data for charging of the MS control speed) are compared with the other methods in the literature, e.g., the JGC-MSCS-CV (JGC-MSCS with constant velocity). JGC-MSCS-NDC (JGC-MSCS with no data collection), the adaptive transmission for RWSN (ATS) [20], the mobile data gathering and energy harvesting in rechargeable WSN (NO-BBA) [11], the data collection in strip-based WSN with mobile data collectors (MDC) [10] in different sensor scales at the same condition setting and time. Figure 8, Figure 9, Figure 10, Figure 11 and Figure 12 show the comparison of obtained results of the proposed method in terms of, e.g., the average network throughput, charging efficiency, packet loss rate, and node energy at the end of the charging with the other methods. The simulation results show that the proposed method of JGC-MSCS can guarantee the charging demand of all nodes while having high network throughput and make the sensor network operate normally.

The computational complexity for the optimization consists of three algorithms: 1 (select bottleneck node), 2 (speed optimization), and 3 (path choice). This means that the computational complexity of the JGC-MSCS algorithm mainly depends on the bottleneck node selection and the relay transmission path selection. The bottleneck node selection problem is to select the “best” charging bottleneck node from all sensor nodes, and then divide the interval according to the speed of the bottleneck node. So, the most expensive equation of the computational complexity is selected to consider the computational complexity that can be determined by selecting the minimum value from the *K*-dimensional vector. Therefore, the worst complexity of the bottleneck node selection scheme is *O(K)*. The problem of relay transmission path selection is to determine the “best” link through two-way variety between ordinary nodes and relay nodes. Similarly, the worst complexity of the relay transmission path selection scheme is, according to Equations (30) and (31), $O\left(K^{2}\right)$.

Figure 8 depicts a comparison of the results of the MS speed controlling strategy of the proposed scheme of JGC-MSCS with the ATS scheme [20] under the setting implementation of the same conditions. The result of the speed control strategy for the rate under seventeen divided allocations in a given strip space area with the *x*-axis start coordinate and the *x*-axis end coordinate. In each region, MS travels at speed V while collecting data and charging. It can be seen from Figure 8 that the proposed JGC-MSCS scheme provides more flexible moving speed controls than the ATS scheme. The obtained moving speed controls by the JGC-MSCS scheme are ascertained according to the hotspot points of the node bottleneck allocation areas. In contrast, the ATS’s speed control always keeps MS running at the maximum or minimum speed.

Figure 9 shows a comparison of the average network throughput of the proposed method with NO-BBA, MDC methods, JGC-MSCS-CV, and JGC-MSCS-NDC with different network node densities. In the scenario for constant moving speed in a strip space with a rate is set to 1 m/s is set to JGC-MSCS-CV. The average network throughput is defined as the sum of the average data sent by all sensor nodes at the end of MS travel. It is clearly seen from Figure 8 that the proposed method (JGC-MSCS) produces the most significant throughput. The traveling speed change of the JGC-MSCS makes reducing the total travel time of round as the concentrated nodes and close to the track. 

Figure 10 displays a comparison of the remaining energy of the node at the end of charging of the proposed method with NO-BBA, JGC-MSCS-CV, and JGC-MSCS-NDC methods for the wireless rechargeable sensor networks. As observed in Figure 9, the proposed method (JGC-MSCS) produces the most significant remaining energy of the node in WRSN.

Table 3 views a comparison of the remaining node power of the proposed JGC-MSCS with the other methods of constant speed traveling, no data collection, and data collection cases. Because of fixed rate of moving approaches make the position of the node to determine the amount of charging that causes the energy imbalance of the node. It is seen that the JGC-MSCS makes the farther sensor nodes get more energy, and the overall charge to the nodes is more, and the mobile charging effect is higher than the competitors.

A parameter is used as the ratio of the total charging capacity of all nodes to the whole time spent traversing the network for measuring charging efficiency.
(34)$$\rho =\frac{\sum^{\;}E_{k}^{total}}{T_{s}}$$
where ρ is a ratio of the total charging capacity of all nodes to the whole time spent traversing. Figure 11 displays the comparison of the charging efficiency of the proposed method for WRSN at different network densities with the JGC-MSCS-CV, JGC-MSCS-NDC, NO-BBA, and MDC methods. It can be seen from Figure 10 that the charging efficiency of the proposed JGC-MSCS has been significantly improved. The speed control of the JGC-MSCS can obtain more energy than the node of the constant speed moving state of the JGC-MSCS-CV under the condition. The remaining power of the nodes under optimization of JGC-MSCS is balanced with a significant charging efficiency.

Figure 12 shows the comparison of the network packet loss rate under different network densities of the proposed JGC-MSCS with the JGC-MSCS-CV, JGC-MSCS-NDC, NO-BBA, and MDC with the number of sensors at 50, 100, 150, and 200, respectively. It can be seen that the rate of packet loss is the smallest amount that belongs to the proposed method of JGC-MSCS. In contrast, without control speed and joined the node data collection algorithm, the JGC-OMSCS-CV scheme caused the highest packet loss rate.

## 5. Conclusions

In this paper, a new solution to maintaining a survival life of the wireless rechargeable sensor networks (WRSN) by controlling the velocity of the mobile sink (MS) charging process was suggested based on analyzing nodes’ energy consumption and residual network energy. A deployed network area as strip space was divided into sub-locations for variant corresponding velocities based on nodes energy expenditure demands. The amount of charging per unit time divided the strip-based space into regions by variant MS speeds and suggests a moving speed control algorithm for MS collecting data and charging nodes in a strip-based area. The points of consumed energy bottleneck nodes in sub-locations are determined based on gathering data of residual energy and expenditure of nodes. A minimum reliable energy balanced spanning tree is constructed based on data collection to optimize the data transmission paths, balance energy consumption, and reduce data loss during transmission. Comparing the experimental results with other methods in the literature show that the proposed scheme offers a more effective alternative in reducing the network packet loss rate, balancing the nodes’ energy consumption, and charging nodes capacity than the competitors. In future work, we plan to investigate, analyze and optimize the path selection of MS under the network model of WRSN in three-dimensional space. The model can introduce optimization with constraints, e.g., obstacles, weather, walking paths and other influencing factors, and even using an unmanned aerial vehicle instead of the MS of the charger.

## Figures and Tables

**Figure 1 sensors-20-04494-f001:**
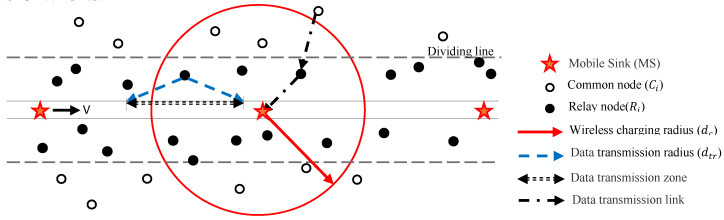
Description of a system model of wireless rechargeable sensor networks (WRSNs).

**Figure 2 sensors-20-04494-f002:**
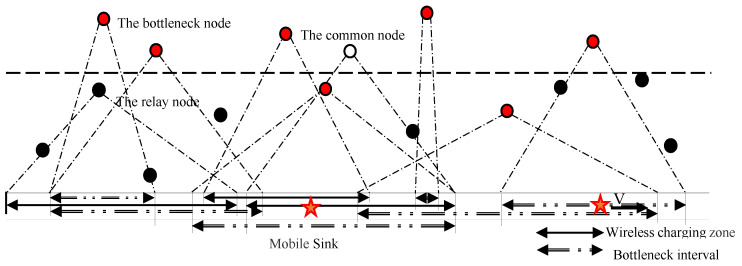
A strip space of WRSN with different regions.

**Figure 3 sensors-20-04494-f003:**
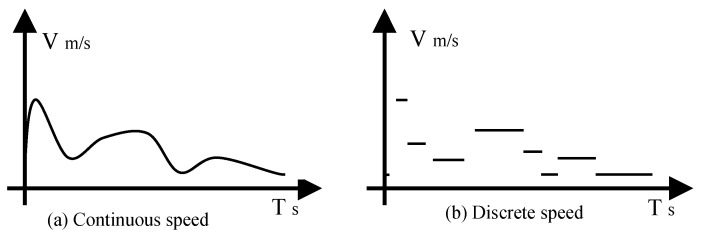
Type of moving speed of mobile sink (MS) in continuous form and converted into a discrete one.

**Figure 4 sensors-20-04494-f004:**
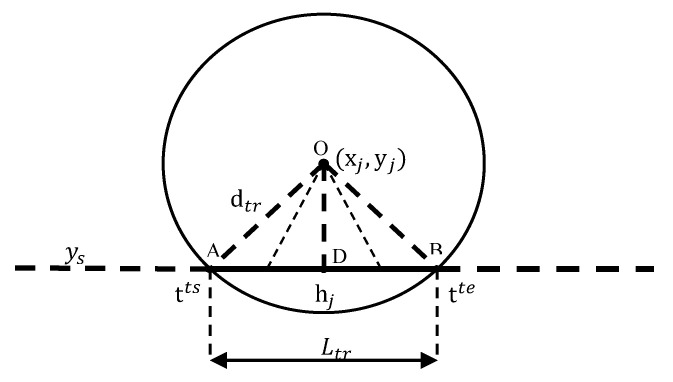
Distance model.

**Figure 5 sensors-20-04494-f005:**
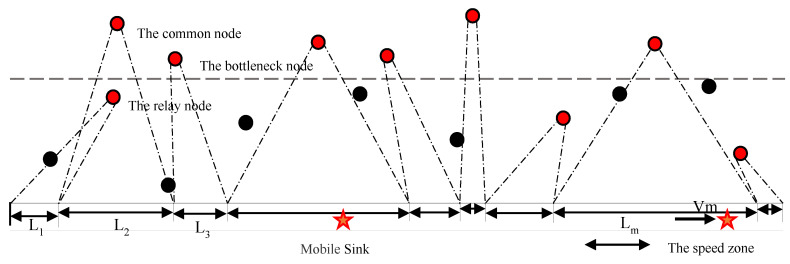
The MS moves on the trajectory according to the area speed.

**Figure 6 sensors-20-04494-f006:**
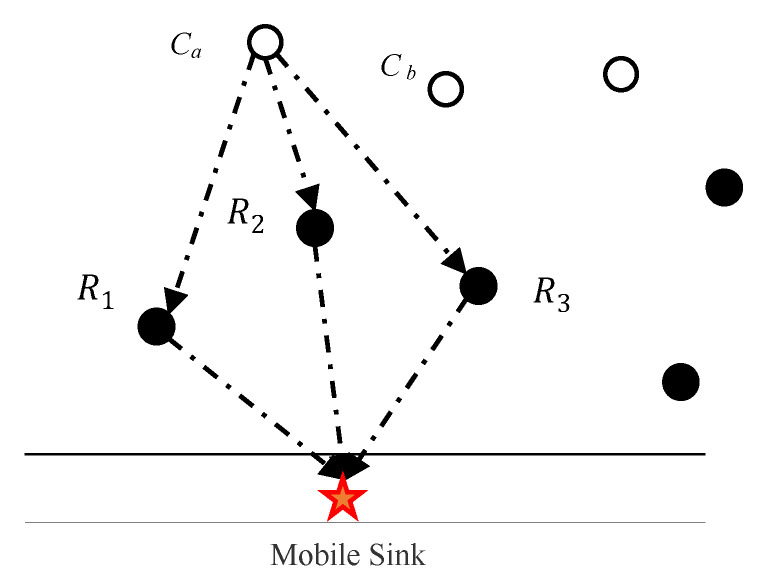
Schematic diagram of transmission path selection.

**Figure 7 sensors-20-04494-f007:**
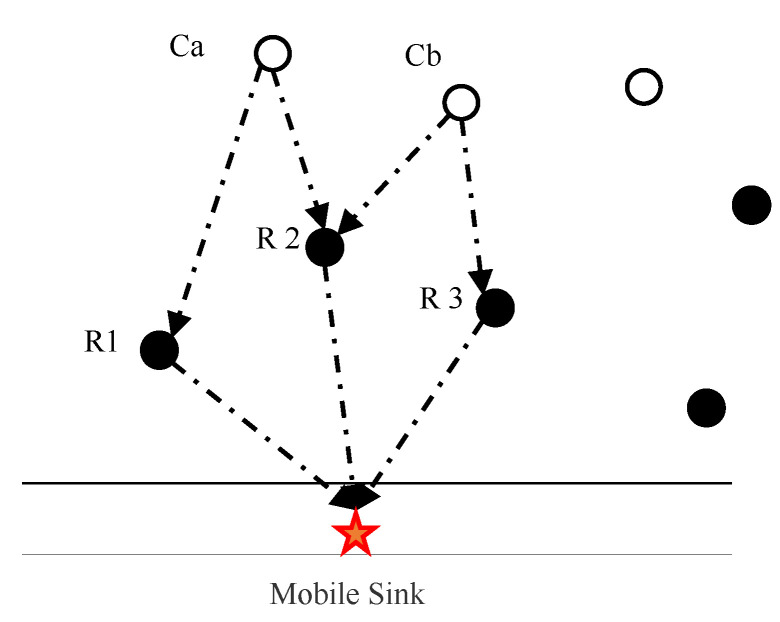
Schematic diagram of bearer selection.

**Figure 8 sensors-20-04494-f008:**
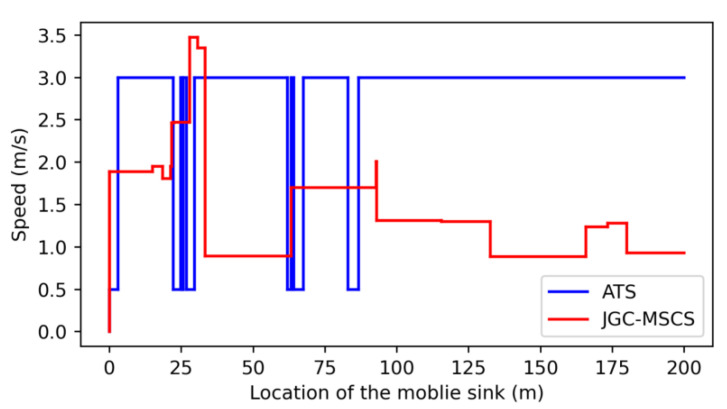
Speed of MS at different locations.

**Figure 9 sensors-20-04494-f009:**
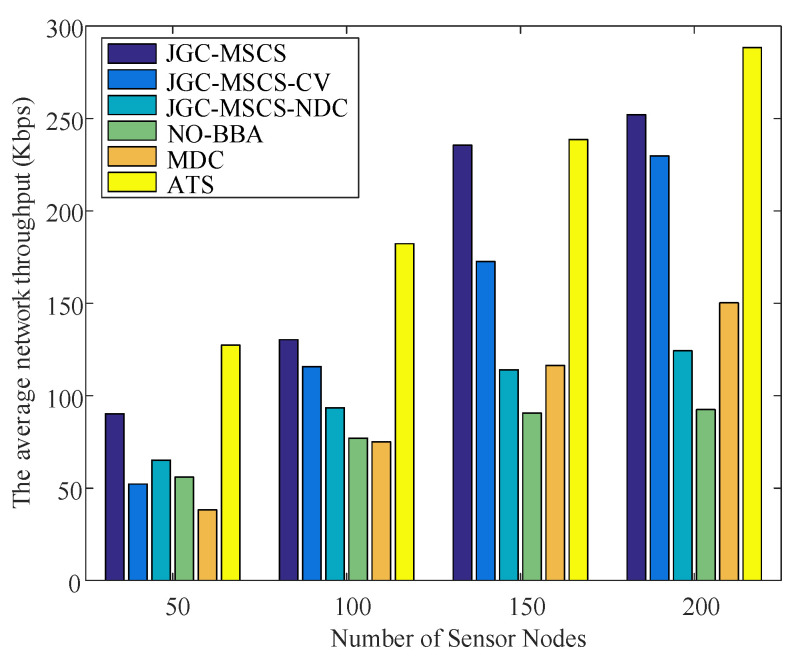
Comparison of the average network throughput.

**Figure 10 sensors-20-04494-f010:**
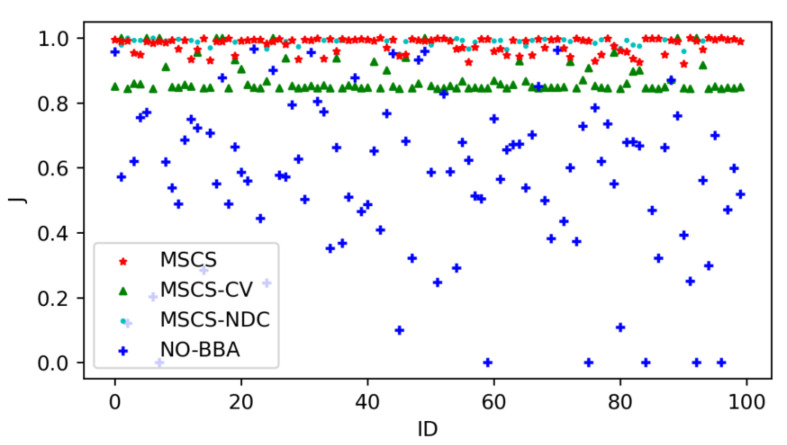
Comparison of the remaining energy of the node.

**Figure 11 sensors-20-04494-f011:**
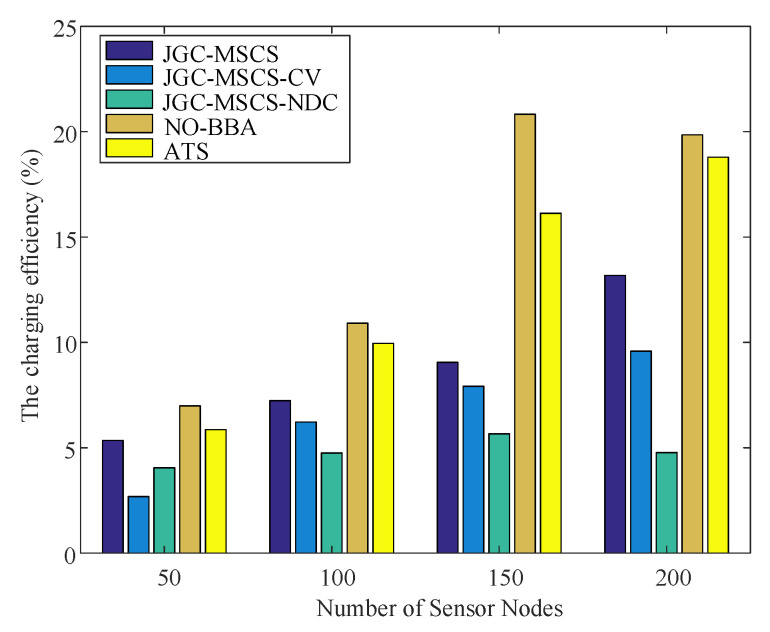
Comparison of the charging efficiency of different network densities.

**Figure 12 sensors-20-04494-f012:**
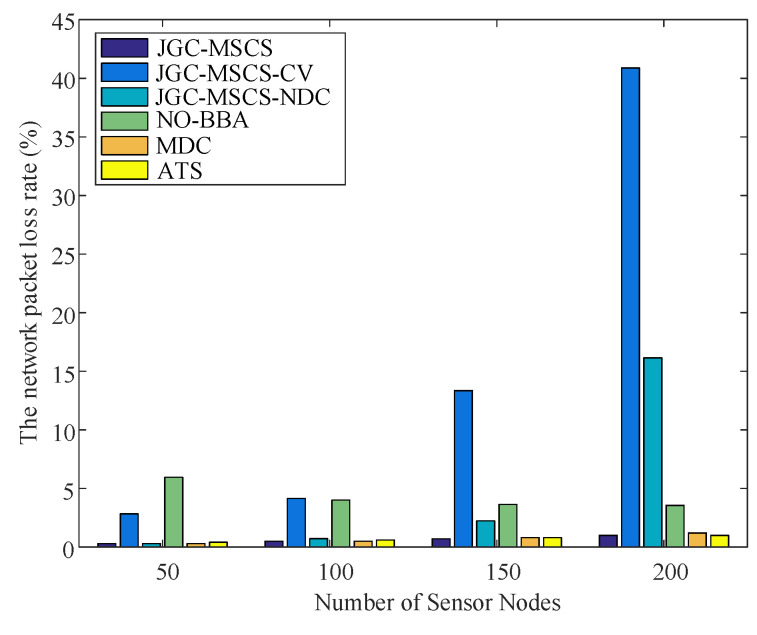
Comparison of the network packet loss rate under different network densities.

**Table 1 sensors-20-04494-t001:** Symbols and definitions.

Symbol	Definition
$B_{full}$	Maximum buffer capacity of the sensor node
$E_{full}$	Maximum battery capacity of the sensor node
R	Set of relay sensor nodes in the one-hop manner
C	Set of common sensor nodes in the multi-hop manner
$B^{init}$	The initial buffer capacity of the sensor node
$E^{init}$	The initial battery capacity of the sensor node
$B_{k}\left(T\right)$	The state of the buffer space of the sensor node at *T*
$E_{k}\left(T\right)$	The state of the battery space of the sensor node at *T*
$d_{tr}$	Transmission radius of the sensor node
$d_{c}$	Charging radius of the sensor node
$d_{ij}$	Distance between node *i* and node *j*
$E_{ij}^{tr}$	Transmission energy consumption of sensor node *i* to node *j*
$E_{ij}^{re}$	Receiving energy consumption of sensor node *j* receiving node *i*
$P_{kr}$	Received charging power of sensor node *k*
$L_{tr}$	Sensor node transmission interval
$t_{j}^{ts}$	Time point of entering the node *j* transmission interval
$t_{j}^{te}$	Time point of leaving the node *j* transmission interval
$t_{k}^{cs}$	Time point of entering the node *k* charging interval
$t_{k}^{ce}$	Time point of leaving the node *k* charging interval
$E_{k}^{consumption}$	Total energy consumption of node *k*
$E_{k}^{charge}$	Total charged energy of node *k*
$G_{j}$	Number of ordinary nodes that relay nodes assist in forwarding data
$E_{kh}^{\prime }$	The amount of charge harvested by node *k* per unit of time
$E_{k}^{helthy}$	Energy threshold of node *k*
$B_{k}^{helthy}$	Buffer threshold of node *k*
$T_{s}$	Total time of mobile charging

**Table 2 sensors-20-04494-t002:** Simulation parameters setting

Definition	Symbol	Value
Full battery capacity	$E_{full}$	1 J
Cache space capacity	$B_{full}$	500 KB
Network length	L	200 m
Network width	W	50 m
Number of nodes	N	100
Transmission radius	$d_{tr}$	20 m
Charging radius	$d_{c}$	30 m
Node acquisition rate	b	512 bits/s

**Table 3 sensors-20-04494-t003:** Comparison table of remaining energy of nodes.

Methods	Mean Deviation	Variance	Standard Deviation	Number of Dead Nodes	Total Travel Time
JGC-MSCS	0.9791 J	0.0005	0.0228	0	167.181 s
JGC-MSCS-CV	0.8758 J	0.0023	0.0486	0	200.001 s
JGC-MSCS- NDC	0.9909 J	0.0001	0.0094	0	283.0963 s
NO-BBA	0.7288 J	0.0344	0.1863	7	200.001 s
MDC	~	~	~	~	269.3333 s

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
