# Peer review of "An Analysis Scheme of Balancing Energy Consumption with Mobile Velocity Control Strategy for Wireless Rechargeable Sensor Networks"

_sensors, 2020, doi:10.3390/s20164494_

Round 1
Reviewer 1 Report
This paper studies maintaining the life cycle of a wireless rechargeable sensor networks (WRSN) by proposing a solution to control the velocity of the mobile sink node charging process based on analyzing energy consumption and residual network energy of its nodes. An optimization problem has been formulated to consider the requirement of the nodes charging energy amount and joint data collection. The performance of the proposed algorithms is evaluated through some comparisons with other approaches from the literature. However, the presentation of the paper is poor in some sections, and it has few typos and grammar errors need to be fixed. Therefore, I cannot recommend the acceptance of the manuscript due to following comments, and suggest to the authors to carefully refine their work and address the listed concerns:
- Comparing with existing work in the open literature, the novelty of the paper and the technical contribution are rather incremental, where some parts of the analysis directly follows from previous works in this area.
- The introduction needs to be improved by considering some relevant references to the studied scenario (not the fixed traveling velocity) and briefly explaining the main differences and the challenges of the mobile gathering and charging.
- In the system model, Figure (1) can be improved by adding the symbols on their respective items/lines in order to ease following the system model with the text and equations. In addition, please differentiate between the data transmission lines in the figure (the dashed ones), it seems they have different shapes but same names.
- Proof of Theorem 1 needs to be presented in an efficient and mathematical way, and pointing the end of the proof.
- The authors provide solutions for the optimization problems. Nonetheless, they do not provide the complexity of the solutions. Thus, a complexity analysis and the inclusion of the results will enhance the practicality of the proposed techniques.
- The presentation of the provided figures in the results section needs to be improved, and make them more visually differentiated.
Author Response
Reviewer#1, Concern # 1:Comparing with existing work in the open literature, the novelty of the paper and the technical contribution are rather incremental, where some parts of the analysis directly follow from previous works in this area.
Author response: The authors much appreciate the constructive comment of the Reviewer#1for the careful reading of the manuscript. Yes, we thank you and agree that compared with existing work in the open literature, the novelty of the paper and the technical contribution are rather incremental.
In our compared results, there are two types of comparisons: options and the other methods. The optional comparisons do with serious in our system that optional control velocity speed and constant speed of velocity, or select join data or without gathering data.
Author action: We updated the manuscript by adding further explanation and cited previous work in Section4, in Figures 8 to 13, and Table 3.
- Experimental Results and Analysis
The obtained experimental results of the proposed method (named JGC-MSCS- Joint and Gather data for charging of the MS control speed) are compared with the other methods in the literature, e.g., the JGC-MSCS-CV (JGC-MSCS with constant velocity). JGC-MSCS-NDC (JGC-MSCS with no data collection), the adaptive transmission for the RWSN with a mobile sink (ATS) [20], the mobile data gathering and energy harvesting in rechargeable WSN (NO-BBA) [9], the data collection in strip-based WSN with mobile data collectors (MDC) [8] in different sensor scales at the same condition setting and time. Figures 8 to 12 show the velocity, the comparison of the average network throughput, charging efficiency, packet loss rate, and node energy at the end of the charging of the proposed method with the other methods. The simulation results show that the proposed method of JGC-MSCS can guarantee the charging demand of all nodes while having high network throughput, and make the sensor network usually operate.
Reference
- Wang, J.; Gao, Y.; Zhou, C.; Sherratt, S.; Wang, L. Optimal coverage multi-path scheduling scheme with multiple mobile sinks for WSNs. Comput. Mater. Contin.2020, 62, 695–711.
- Lan, X.; Zhang, Y.; Cai, L.; Chen, Q. Adaptive Transmission Design for Rechargeable Wireless Sensor Network with a Mobile Sink. IEEE Internet Things J.2020, 1.DOI: 10.1109/JIOT.2020.3001034
Reviewer#1, Concern # 2:The introduction needs to be improved by considering some relevant references to the studied scenario (not the fixed traveling velocity) and briefly explaining the main differences and the challenges of the mobile gathering and charging.
Author response: The authors much appreciate the constructive comment of the Reviewer#1. We have added descriptions as the suggestion of the comment and cited some works of literature.
Author action: We updated the manuscript by adding further explanation and cited previous work for Section 1 as follows:
However, most of these mentioned charging methods applied to the fixed traveling velocity of MS as a constant that made the MS spending more time on the trajectory [7]. Especially in a narrow and long space, and a running MS may overflow the charge of some nodes that caused an unbalanced phenomenon and not an efficiency charger. The nodes in WRSN with high energy consumption, generally because of the need to forward a large amount of data to other nodes, or the distance of transmitted data packets is relatively long, requires more energy from charging capacity than the other nodes with fewer load tasks and low energy consumption[15].
Moreover, there are a few previous works that also considered changing the driving velocity speed of the mobile charger to get better the charging ability of the portable charger [16][17]. The distances between the charging points and the residence time of the charging points were regarded to adjust the moving speed on the path and reducing the charging completion time. These methods have advantages of determining the optimal rate of the mobile data receiver and avoiding the energy hole problem by analyzing transmission energy consumption.
However, the bottlenecks points of node energy consumption and the data transmission paths optimization have been not considered seriously for the core issue of how to control the charger's moving speed. So the energy consumption of the cluster head node (its place where is much higher load than that of other nodes) would die faster caused the entire network reduced performance.
This paper considers adjusting the velocity for the MS moving based on optimization of the requirement of the nodes charging energy amount and joint data collection for maintains the regular operation of the network. The highlights of the contribution in this paper are listed as follows.
- Analyzing the meaning of the MS velocity speed control problem in the WRSN through determining the points of the bottlenecks node energy consumptions with the higher energy consumption nodes in WSN is the more requirements energy from the charging amount.
- Optimizing the data transmission paths by constructing reliable energy balanced spanning trees based on data collection and residual energy network.
- Suggesting a moving speed control scheme for MS collecting data and charging nodes in a strip-based area according to the optimal demand of the node to extend the life of the network.
- Implementing many experiments to verify the reliability of the proposed scheme for the MS speed control through a two-distinction routing graph for charging and collecting data to avoid hot spots and reduce the data packet loss rate.
In the References part:
- Chen, F.; Zhao, Z.; Min, G.; Gao, W.; Chen, J.; Duan, H.; Yang, P. Speed control of mobile chargers serving wireless rechargeable networks. Futur. Gener. Comput. Syst.2018, 80, 242–249.
- Lan, X.; Zhang, Y.; Cai, L.; Chen, Q. Adaptive Transmission Design for Rechargeable Wireless Sensor Network with a Mobile Sink. IEEE Internet Things J.2020, 1.DOI: 10.1109/JIOT.2020.3001034
Reviewer#1, Concern # 3:In the system model, Figure (1) can be improved by adding the symbols on their respective items/lines in order to ease following the system model with the text and equations. In addition, please differentiate between the data transmission lines in the figure (the dashed ones), it seems they have different shapes but same names.
Author response: The authors are much thankful for the constructive comment of Reviewer#1. Yes, we have revised this Figure 1 and also all other Figures.
Author action: We updated the manuscript by adding further explanation and for Figure 1.
Reviewer#1, Concern # 4:Proof of Theorem 1 needs to be presented in an efficient and mathematical way, and pointing the end of the proof.
Author response: The authors are much thankful for the valuable comment of Reviewer#1. We have proved Theorem 1 using formula derivation.
Author action: We updated the manuscript by adding further explanation and providingproof in section3.1
3.1 System Analysis
Refer to Eq.(9), if MS moves slowly (denoted ), the charged node's amount ( will be got more fully.
………
Theorem 1: For all sensor nodes within the MS charging range, the time required for node charging is , . If all nodes are charged with the charging time of node Si, and is the longest time, then all sensor nodes can eventually reach a fully charged state. We also provide "Proof 1" with more details.
Reviewer#1, Concern # 5:The authors provide solutions for the optimization problems. Nonetheless, they do not provide the complexity of the solutions. Thus, a complexity analysis and the inclusion of the results will enhance the practicality of the proposed techniques.
Author response: The authors much appreciate the constructive comment of the Reviewer#1.
Author action: We updated the manuscript by adding a further explanation of a complexity analysis in section4 as follows.
- Experimental Results and Analysis
Assumed sensor network deployment with N rechargeable sensors……
The computational complexity for the optimization consists of three algorithms 1 (select bottleneck node), 2 (speed optimization), and 3 (path choice). It means that short name as the JGC-MSCS algorithm mainly depends on the computational complexity of the bottleneck node selection problem and the relay transmission path selection problem. The bottleneck node selection problem is to select the "best" charging bottleneck node from all sensor nodes, and then divide the interval according to the speed of the bottleneck node. The most expensive equation of the computational complexity is selected to consider the computational complexity that can be determined by selecting the minimum value from the K-dimensional vector. Therefore, the worst complexity of the bottleneck node selection scheme is O(K). The problem of relay transmission path selection is to determine the "best" link through two-way variety between ordinary nodes and relay nodes. Similarly, the worst complexity of the relay transmission path selection scheme as according to Eqs(30) and (31) that is .
Reviewer#1, Concern # 6:The presentation of the provided figures in the results section needs to be improved, and make them more visually differentiated.
Author response: The authors are grateful for the valuable comment of the Reviewer#1. We have adopted according to the suggestion and improved the presentation of the provided figures in Section 4.
Author action: We updated the manuscript by revising Figures in Section 4.
Reviewer 2 Report
This paper proposes a scheme wireless charging and data collection in a WRSN with mobile sink (MS) where the scheme looks at controlling the speed of the MS to optimize the performance. The paper is well structures and the discussion is well supported by results and benchmarking. However the English language is very poor and often the meaning of the sentence is lost.
The authors claim that this is the first work that models the velocity control (both speed and path) in a WRSN scheme, however we found the following recent work that models the MS speed, albeit differently:
X. Lan, Y. Zhang, L. Cai and Q. Chen, "Adaptive Transmission Design for Rechargeable Wireless Sensor Network with a Mobile Sink," in IEEE Internet of Things Journal, doi: 10.1109/JIOT.2020.3001034.
Given that the authors main contribution claimed by the authors is around the modeling of the MS velocity and impact on energy consumption and given that the above work proposes a similar scheme, it is essential for the authors to compare their proposed scheme to the one by Lan et.al above to establish the level of contribution in this work.
Author Response
- Reviewer#2, Concern # 1:The paper is well structures and the discussion is well supported by results and benchmarking. However, the English language is very poor and often the meaning of the sentence is lost.
Author response: The authors much appreciate Reviewer#2 for the careful reading of the manuscript. We have tried to correct all typos, spelling, and grammar in the whole document.
Author action: We updated the manuscript by correcting spelling and grammar in the entire paper.
- Reviewer#2, Concern # 2:The authors claim that this is the first work that models the velocity control (both speed and path) in a WRSN scheme, however we found the following recent work that models the MS speed, albeit differently:
[20] X. Lan, Y. Zhang, L. Cai and Q. Chen, "Adaptive Transmission Design for Rechargeable Wireless Sensor Network with a Mobile Sink," in IEEE Internet of Things Journal, doi: 10.1109/JIOT.2020.3001034.
Given that the authors main contribution claimed by the authors is around the modeling of the MS velocity and impact on energy consumption and given that the above work proposes a similar scheme, it is essential for the authors to compare their proposed scheme to the one by Lan et.al above to establish the level of contribution in this work.
Author response: Yes, the suggested reference [20] has just published online.The authors are thankful for the valuable comment of the Reviewer#2. We have revised according to your suggestion and added the presentation of explain and figure 8 in section 4.
Author action: We updated the manuscript by revising of compared result curve controlling in Figure 8, the presentation of the comparison, and explanation in Section 4.
Figure 8 illustrates a comparison of the results of the MS speed controlling strategy of the proposed scheme of JGC-MSCS with the ATS scheme[20]. The result of the speed control strategy for the rate under seventeen divided allocations in a given strip space area with the x-axis start coordinate and the x-axis end coordinate. In each region, MS travels at speed V while collecting data and charging tasks of the sink. Observed from Figure 8, the proposed JGC-MSCS scheme provides the moving speed controls are more flexible than the ATS scheme. The obtained moving speed controls by the JGC-MSCS scheme go drawly according to the hotspots points of the node bottleneck allocation areas. In contrast, the AST's speed control always keeps MS running at the maximum or minimum speed.
References section:
- Lan, X.; Zhang, Y.; Cai, L.; Chen, Q. Adaptive Transmission Design for Rechargeable Wireless Sensor Network with a Mobile Sink. IEEE Internet Things J.2020, 1.DOI: 10.1109/JIOT.2020.3001034
Reviewer 3 Report
With the aim to maximize the life cycle of the sensor networks, authors present a theoretical and experimental study about of a novel strategy for mobile charging of sensor networks based on controlling the moving speed of the mobile charger. Essentially, to ensure the charging demand of any node, authors discretize the strip space of Wireless Rechargeable Sensor Networks (WRSN) into different regions by identifying the bottleneck nodes, and then adjusting the moving speed of the mobile charger, as well as charging rate in each section. Both, experimental and theoretical results seem to be very promising when compared with existing and former mobile charging strategies based on constant moving speed. Although, authors assumed some ideal conditions for solving the model, it is clear that in actual WRSN surrounding conditions could affect the performance of their proposal, for instance:
- Although, It is not specified clearly the type of mobile vehicle that could be used for transporting the charging system (from the manuscript can be inferred a car), how can be the traffic conditions affected the performance of the present platform for mobile charging?, the discretization time in this event can be very large; so, how will it affect the best path selection?, and how will it affect the speed control?.
- In the event aforementioned, can be your algorithms adapted to anticipate; for instance, the traffic conditions or even weather conditions?; this with the aim of optimizing energy consumption.
- Could be used a Drone as mobile vehicle?, it might avoid traffic conditions.
- The transmission power given by the charging model (equation-4 in the manuscript) is inversely proportional to the square of the distance between the node and the mobile sink. Clearly such distance is variable, and it depends on time-T as you correctly specify. However, according to equation-7 you have taken only into consideration two coordinates x-y, in actual WRSN sensors are distributed in the space x-y-z, and possibly both x(T) and z(T) depend on time T. Would be possible to extend your results into a 3D space sensors distribution so as to get a more applicable model?. Even, by using a Drone as mobile vehicle the three spatial variables x-y-z will depend on time, x(T), y(T) and z(T). It will improve too much your model.
- According to your results, the speed of the mobile vehicle must be approximately between 0.88m/s and 3.47m/s; it is a very low speed for any typical mobile vehicle as a car. Considering that, in average people tend to walk at about 1.4m/s, what would be an adequate mobile vehicle for using it as mobile charger?
- Besides, although paper is well English written, but typo errors were identified along the manuscript; for instance, page 4 row 115 in the phrase “and he transmission power”. Please English double check.
In spite of the assumptions, theoretical results seem to be very promising; they give an important contribution so as to design the future mobile charger systems for wireless sensor networks. Hence, based on the aforementioned observations, the present research cannot be accepted as it is, it needs a major revision before being publishable.

Author Response
- Reviewer#3, Concern # 1:: …can be your algorithms adapted to anticipate; for instance, the traffic conditions or even weather conditions?; this with the aim of optimizing energy consumption. Could be used a Drone as mobile vehicle?, it might avoid traffic conditions….,
Author response: The authors much are thankful for a rising good question of the comment.That is an excellent suggestion about the further constraints for the optimization problem of the path selection. However, this paper only focused on how to maintain the sensor network by efficiently charging its nodes battery. The weather conditions or using Drone have not been taken into account in our paper. We hope it will be as the future work and added in the conclusion section.
Author action: We updated the manuscript by putting this as our future work in the conclusion part.
- Conclusion
In this paper, a new solution to maintaining a survival life of the wireless rechargeable sensor networks (WRSN) by controlling the velocity of the mobile sink node (MS) charging process was suggested based on analyzing nodes' energy consumption and residual network energy. A deployed network area as strip space was divided into sublocations for variant corresponding velocities based on nodes energy expenditure demands. The amount of charging per unit time divided the strip-based space into regions by variant MS speeds and suggests a moving speed control algorithm for MS collecting data and charging nodes in a strip-based area. The points of consumed energy bottleneck nodes in sublocations are determined based on gathering data of residual energy and expenditure of nodes. A minimum reliable energy balanced spanning tree constructed based on data collection to optimize the data transmission paths, balance energy consumption, and reduce data loss during transmission. The compared experimental results with the other methods in the literature show that the proposed scheme offers a more effective alternative in reducing the network packet loss rate, balancing the nodes' energy consumption, and charging nodes capacity than the competitors. As future work, the network model of WSN with three-dimensional space will be considered to analyze and optimize the path selection of moving MS. The model can be introducing optimization with constraints, e.g., obstacles, weather, walking paths and other influencing factors, and even using aircraft instead of an MS of the charger.
- Reviewer#3, Concern # 2:Page 4 row 115 in the phrase "and he transmission power.Please English double check!
Author response: The authors much appreciate Reviewer#3 for the careful reading of the manuscript. We
have tried to correct all typos, spelling, and grammar in the whole manuscript.
Author action: We updated the manuscript by correcting spelling and grammar in the entire paperto make sure there were no mistakes.
Round 2
Reviewer 1 Report
The authors have appropriately addressed all my concerns from the previous round of review and made the necessary modifications to the manuscript. However, there are still some minor writing issues in the manuscript need to be fixed, especially in the reference list.
Author Response
Q:The authors have appropriately addressed all my concerns from the previous round of review and made the necessary modifications to the manuscript. However, there are still some minor writing issues in the manuscript need to be fixed, especially in the reference list.
A: Many thanks for your valuable comments. We have once again carefully proofread this paper, with special attention to the reference format. As can be seen from the 2nd round revised manuscript.
Once again, many thanks!
Reviewer 2 Report
The authors have addressed the main concern related to the comparison with the state of the art research highlighted in my previews review. Their results highlight clearly the technical contribution and the gains of the proposed method
The English language is extremely poor and not unintelligible in some places (e.g., the added sentences in Section 4).
Author Response
Q: The authors have addressed the main concern related to the comparison with the state of the art research highlighted in my previews review. Their results highlight clearly the technical contribution and the gains of the proposed method.
The English language is extremely poor and not unintelligible in some places (e.g., the added sentences in Section 4).
A: Many thanks for your valuable comments. We have once again carefully proofread this paper, with special attention to the added sentences in Section 4. As can be seen from the 2nd round revised manuscript.
Once again, many thanks to your kind suggestion!
Reviewer 3 Report
I would like comment that based on responses given by the authors to questions; they have modified the main manuscript by correcting sentences and graphs which complement adequately the discussions of their results. In this sense, we believe that the research contains important data and information which could be relevant to the literature and scientific community. Therefore, I recommend accepting the article for publication in the present form.